# Bacterial community dynamics are linked to patterns of coral heat tolerance

Maren Ziegler[1], Francois O. Seneca[2], Lauren K. Yum[1], Stephen R. Palumbi[2] & Christian R. Voolstra[1]

Ocean warming threatens corals and the coral reef ecosystem. Nevertheless, corals can be adapted to their thermal environment and inherit heat tolerance across generations. In addition, the diverse microbes that associate with corals have the capacity for more rapid change, potentially aiding the adaptation of long-lived corals. Here, we show that the microbiome of reef corals is different across thermally variable habitats and changes over time when corals are reciprocally transplanted. Exposing these corals to thermal bleaching conditions changes the microbiome for heat-sensitive corals, but not for heat-tolerant corals growing in habitats with natural high heat extremes. Importantly, particular bacterial taxa predict the coral host response in a short-term heat stress experiment. Such associations could result from parallel responses of the coral and the microbial community to living at high natural temperatures. A competing hypothesis is that the microbial community and coral heat tolerance are causally linked.

[1] Red Sea Research Center, Division of Biological and Environmental Science and Engineering (BESE), King Abdullah University of Science and Technology (KAUST), Building 2, Thuwal 23955-6900, Saudi Arabia. [2] Hopkins Marine Station, Stanford University, 120 Ocean View Blvd, Pacific Grove, California 93950, USA. Correspondence and requests for materials should be addressed to S.R.P. (email: spalumbi@stanford.edu) or to C.R.V. (email: christian.voolstra@kaust.edu.sa).

Bacterial microorganisms play important roles in shaping animal biology in many systems[1–3], emphasizing the diversity and functional capacity of bacteria, and challenging our views on what constitutes a genome or an organism[2,4]. In particular, long-lived sessile stony corals are dependent on an endosymbiosis with photosynthetic algal symbionts in the genus Symbiodinium[5] and, more recently, are shown to associate with a diverse set of bacteria that contribute important functions to the coral host[6–8]. The coral animal and associated organisms together comprise the coral holobiont[9]. The composition of the coral holobiont varies across environments, which may entail coral host genomic[10] and Symbiodinium community[11,12] changes that influence the resilience of the coral holobiont to environmental stressors. Consequently, changes in the bacterial community also represent an opportunity for a yet unexplored source of organismal adaptation that may occur within the reproductive lifetime of the host[7,13,14]. Yet, it remains unknown to what extent the microbial community may contribute to coral host resilience in a changing environment, for example, under ocean warming.

Understanding and disentangling how the coral host and symbiont compartments interact and are affected by environmental change require a study system. Such a system can be found in the coral Acropora hyacinthus in the back reef pools of Ofu Island of the U.S. National Park of American Samoa. These back reef pools feature thermally distinct environments in direct vicinity that allow exploring the effects of the environment on coral holobionts without the confounding factor of site. In addition, A. hyacinthus is a thermally sensitive cosmopolitan species with high prevalence on Pacific reefs[15]. Previous studies on A. hyacinthus in the back reef pools of Ofu Island could show that Symbiodinium communities differ between corals from two thermally distinct environments (i.e., a highly variable HV pool and a moderately variable MV pool), although under heat stress MV corals bleached irrespective of which symbiont type they harboured[16]. This has been explained in part by contrasting genomic adaptation between corals from the HV and MV pool[10,17], but the data also show strong acclimatization between the thermal environments after reciprocal transplantation[18]. Hence, acclimatization of individual colonies after transplantation was not due to changes in intracellular symbiont genotypes[18], but could have involved changes in other symbionts or associated microbes. In line with this, recent work has shown strong association of coral colonies with species-specific communities of typically 100s of bacterial taxa[6,19], some of which might mediate sensitivity to environmental variation such as nutrient concentrations or perhaps temperature[13].

Given the importance of bacteria to holobiont function[1–3], we sought to investigate bacterial community composition and its potential role in contributing to thermal resistance of the coral holobiont. To do this, we conducted a long-term reciprocal transplantation experiment of A. hyacinthus corals at Ofu Island between the HV and MV pools, which experience strong differences in their absolute and daily temperature regimes, and assessed bacterial community changes as a result of transplantation to different environments. Subsequently, we assessed bacterial dynamics in a short-term heat stress experiment and how they differed as a result of prior residence in different thermal habitats.

Our study shows that bacterial community composition and underlying functional profiles associated with reef corals are different across thermally variable habitats and adapt to the new environment when corals are reciprocally transplanted. Subsequent exposure of these corals to thermal bleaching conditions changes the microbial community of heat-sensitive corals coming from a more stable, cooler environment. In contrast, heat-tolerant corals from a highly variable, warmer environment frequently exposed to heat stress harbour a stable microbial community and bleached less. This effect is irrespective of the origin of the corals and the coral genotype, but rather determined by the environment that transplanted corals where exposed to prior to the short-term heat stress experiment.

## Results

**Coral microbiomes differ with thermal habitat.** During our 17-months transplantation experiment, A. hyacinthus corals were exposed to contrasting thermal environments in the back reef pools of Ofu Island. In the MV pool, temperatures range between 26 and 33 °C, but rarely exceed 32 °C, while in the HV pool, temperatures are more extreme, fluctuating between 25 and 35 °C, and regularly exceeding the local bleaching threshold of 30 °C (ref. 16) (Fig. 1). In each pool, at least eight colonies were sampled and fragmented into eight pieces each to control for host genotype. Subsequently, half of the fragments were transplanted back to the pool of origin and the other half was cross-transplanted to the non-native pool (Fig. 1a). To assess differences in coral bacterial community composition, we used RNA-based amplicon sequencing of the 16S rRNA marker gene (Supplementary Data 1).

Microbial community composition differed between coral colonies in the thermal environments of the HV and MV pools (ANalysis Of SIMilarity (ANOSIM), $R = 0.122$, $P < 0.001$). The most abundant bacterial families that were detected in corals in the HV pool were Alteromonadaceae and Rhodospirillaceae (each about 15%), followed by Hahellaceae (13%; annotated as Endozoicomonaceae in the Greengenes Database[20]). In corals in the MV pool, Hahellaceae (27%) and Alteromonadaceae (24%) were also abundant, but Rhodospirillaceae accounted for less than 2% and were pronouncedly less abundant. Other bacterial families that were also significantly less abundant in corals in the MV pool were Flammeovirgaceae, Piscirickettsiaceae, Hyphomicrobiaceae, Phycisphaeraceae, and Spirochaetaceae (Fig. 1d).

**Coral microbiomes adapt to thermal habitats.** Reciprocal transplantation of coral colonies between the two thermally distinct environments strikingly demonstrated that coral microbial communities fully adjusted to the new environmental conditions: 17 months after transplantation, the microbiomes of non-native corals transplanted into each pool could not be distinguished from the microbiomes of the native corals in the same pool (Fig. 1c, ANOSIM, HV pool $R = 0.013$, MV pool $R = 0.005$, both $P > 0.05$). At the same time, cross-transplanted fragments were significantly different from their back-transplanted (genet) counterparts (Fig. 1c, ANOSIM, HV pool $R = 0.149$, $P < 0.005$, MV pool $R = 0.184$, $P < 0.001$).

To test for co-variation of coral host genotypes and bacterial symbionts across environments that would point towards a relationship between host genetic variance and bacterial association, we tested whether distinct genotypes of A. hyacinthus were associated with distinct bacterial OTUs (Operational Taxonomic Units). We did not find any particular bacterial taxon associated with any particular coral colony (genotype), and hence, our study suggests a lack of covariance between coral genotype and microbiome composition. The apparent lack of covariance requires further attention and argues against a heritable microbial component of the coral holobiont in this species. Our data may thus contribute important insights to the debate around the hologenome theory and further studies are warranted that investigate in detail the relationship between animal or plant hosts and associated bacterial symbionts[14].

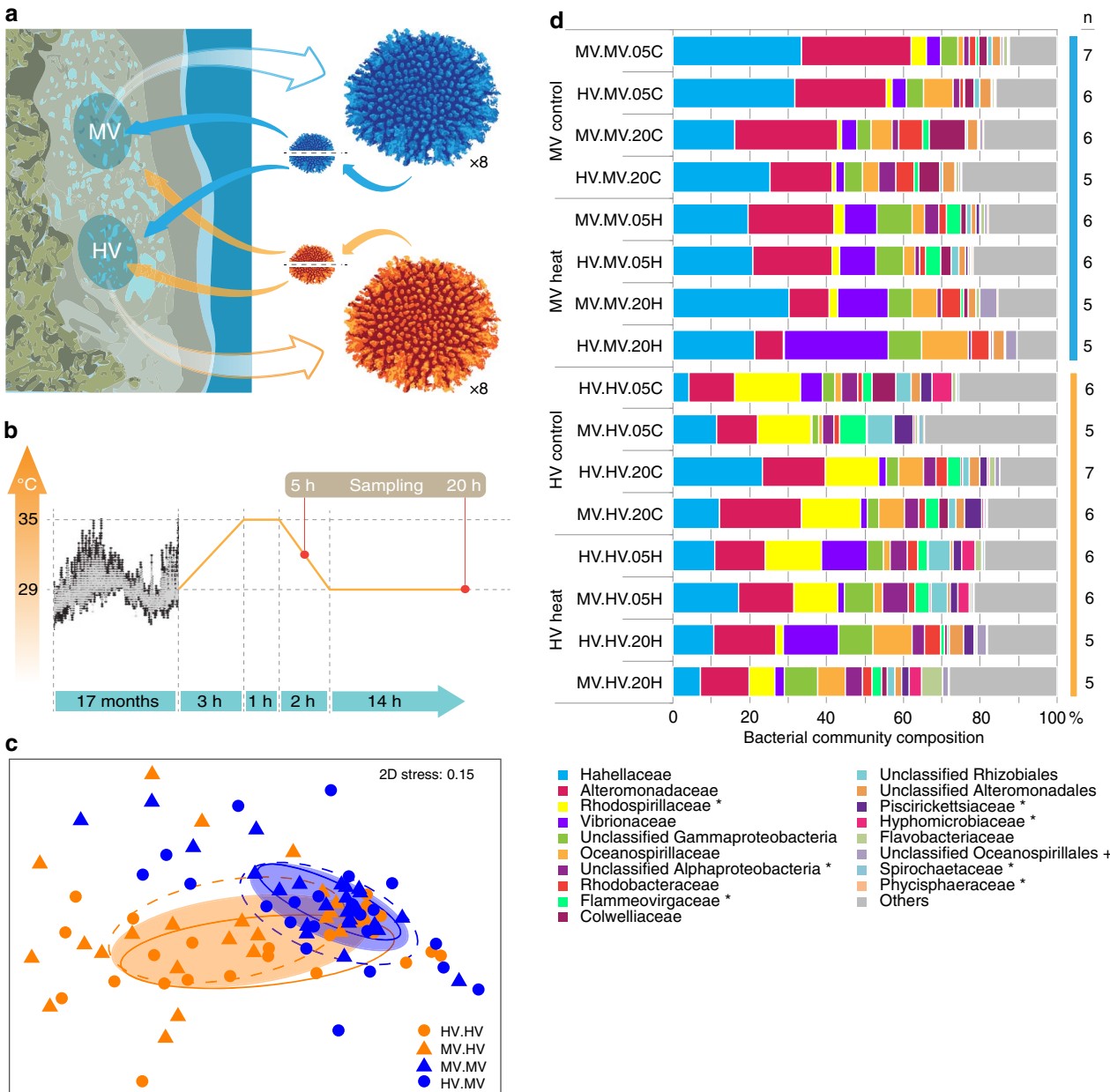

**Figure 1 | Long-term transplantation and short-term heat stress experiment.** (**a**) View of the back reef pool locations (HV and MV pool) on American Samoa, and sampling design of the reciprocal transplantation experiment of fragmented colonies of *Acropora hyacinthus* between the pools; (**b**) temperature profiles measured in the HV (black) and MV (grey) pools over time (modified following Oliver and Palumbi[24]) and short-term heat stress experiment and sampling; (**c**) non-metric Multidimensional Scaling (nMDS) of bacterial community composition after transplantation between HV and MV pools; each symbol represents a sample, symbol shapes denote pool of origin (circles = HV pool, triangles = MV pool), symbol colours denote pool of destination (orange = HV pool, blue = MV pool), ellipses are drawn around each group's centroid; ellipse lines: dashed for cross-transplants, continuous for back-transplants, filled ellipse for significant pool-of-destination groups (ANOSIM, $R = 0.122$, $P < 0.001$); (**d**) 16S rRNA sequence-based microbial community composition of reciprocally transplanted colonies between HV and MV pool locations on the bacterial family level. Differentially abundant bacterial families between the two pools are marked with an '*' if more abundant in the HV pool and with a '+' if more abundant in the MV pool. HV, highly variable pool, MV, moderately variable pool, 05C and 20C = 5 and 20 h control (C) short-term heat stress experiment, 05H and 20H = 5 and 20 h treatment (H) short-term heat stress experiment. Sample name scheme in (**c**,**d**): 'pool-of-origin.pool-of-destination.time (05 = 5 h, 20 = 20 h) treatment (C = control, H = heat)' in short-term heat stress experiment.

Based on the complete transplantation effect of the coral-associated bacterial communities (i.e., corals in the same pool harbour the same microbiome irrespective of their pool of origin), colonies are referred to as HV and MV corals according to their pool of destination in the following. Data from the long-term reciprocal transplantation also allowed the differentiation between a fixed, shared microbiome present in

all coral colonies and a more flexible, acquired microbiome resulting from exposure to contrasting thermal environments (Fig. 1). For this, we followed the approach taken by Hernandez-Agreda *et al.*[21] and partitioned the bacterial microbiome into a ubiquitous core microbiome of few symbiotic host-selected bacteria (Supplementary Data 2), a microbiome of spatially explicit core taxa (i.e., environmentally linked microbes)

**Table 1 | Enrichment of protein functions in coral microbiomes from different thermal habitats.**

| Functional annotation (Enzyme Commission number) | KEGG | LDA mean | Enrichment | LDA group mean | P-value |
|---|---|---|---|---|---|
| Primary amine oxidase (EC 1.4.3.21) | K00276 | 1.893 | HV | 1.605 | 1.57E-08 |
| Electron transfer flavoprotein quinone-oxidoreductase (EC 1.5.5) | K00313 | 1.953 | HV | 1.637 | 8.26E-09 |
| Sarcosine dehydrogenase (EC 1.5.99.1) | K00314 | 2.372 | HV | 1.812 | 5.24E-08 |
| Phosphatidyl N-methylethanolamine N-methyltransferase (EC 2.1.1.71) | K00570 | 1.896 | HV | 1.842 | 7.19E-09 |
| Erythritolkinase (EC 2.7.1.27) | K00862 | 1.890 | HV | 1.855 | 1.19E-08 |
| Acetyl ornithine deacetylase (EC 3.5.1.16) | K01438 | 2.720 | HV | 1.537 | 2.01E-05 |
| Adenylatecyclase (EC 4.6.1.1) | K01768 | 2.969 | HV | 2.300 | 7.12E-08 |
| Multiple sugar transport system ATP-binding protein | K02023 | 3.309 | HV | 2.571 | 8.33E-07 |
| Putative spermidine putrescine transport system ATP-binding protein | K02052 | 3.171 | HV | 2.514 | 1.05E-07 |
| Simple sugar transport system ATP-binding protein (EC 3.6.3.17) | K02056 | 3.279 | HV | 2.593 | 1.42E-07 |
| Simple sugar transport system substrate-binding protein | K02058 | 3.246 | HV | 2.560 | 1.90E-07 |
| L-fucose mutarotase (EC 5.1.3) | K02431 | 2.208 | HV | 1.674 | 3.69E-08 |
| Nitrogen-fixation protein NifW | K02595 | 1.926 | HV | 1.557 | 9.49E-09 |
| Ferredoxin-like protein | K03855 | 1.953 | HV | 1.634 | 8.26E-09 |
| Chaperonin GroES | K04078 | 2.919 | HV | 1.970 | 8.66E-09 |
| Dihydroxy-acetonekinase N-terminal domain (EC 2.7.1) | K05878 | 2.189 | HV | 1.677 | 3.09E-08 |
| Anthraniloyl CoA monooxygenase (EC 1.14.13.40) | K09461 | 1.987 | HV | 1.504 | 9.63E-08 |
| Ribose transport system ATP-binding protein (EC 3.6.3.17) | K10441 | 2.836 | HV | 2.113 | 7.76E-08 |
| Fructose transport system substrate-binding protein | K10552 | 1.937 | HV | 1.939 | 7.43E-08 |
| Fructose transport system permease protein | K10553 | 1.937 | HV | 1.937 | 6.82E-08 |
| Fructose transport system ATP-binding protein | K10554 | 2.015 | HV | 1.659 | 8.11E-08 |
| DtxR-family transcriptional regulator manganese-transport regulator | K11924 | 2.038 | HV | 1.553 | 1.80E-08 |
| Uncharacterized oxidoreductase (EC 1.1.1) | K13574 | 2.152 | HV | 1.695 | 6.82E-08 |
| Formyl tetrahydrofolate deformylase (EC 3.5.1.10) | K01433 | 2.624 | MV | 1.813 | 3.53E-08 |
| Exonuclease SbcD | K03547 | 2.413 | MV | 1.690 | 5.14E-07 |
| Phosphate acetyl-transferase (EC 2.3.1.8) | K13788 | 2.363 | MV | 1.742 | 1.30E-07 |

Microbial communities in heat-tolerant corals from the highly variable (HV) pool and heat-sensitive corals from the moderately variable (MV) pool are characterized by differentially enriched proteins. Linear discriminant (LDA) effect size was used to test the enrichment of individual functions annotated as KEGG Orthologs. For each function, the LDA group mean and P-value are listed for the enriched pool, respectively.

(Supplementary Data 3), and transient, loosely associated bacteria (remaining OTUs, Supplementary Data 1).

**Microbiome dynamics are linked to coral heat tolerance**. To understand the microbial response to heat stress in coral colonies from different thermal environments, we subjected coral colony fragments to a short-term simulated bleaching experiment. The temperature profile during the experimental treatment mimicked the natural temperature variation in the HV pool during summer noon low tides[16] that cause coral bleaching. Corals in the heat stress treatment were exposed to increasing temperatures from 29 to 35 °C over 3 h, held at 35 °C for 1 h, and returned to 29 °C within 2 h, while the control temperature was kept at a constant 29 °C (Fig. 1b). Samples were collected after 5 h to capture the early onset of heat stress and again after 20 h to coincide with the onset of visually detected bleaching[22].

Coral-associated microbial communities responded to short-term heat stress within 20 h (ANOSIM, R 0.093, P < 0.001; Fig. 1). However, the thermal environment in which the corals spent 17 months prior to short-term heat exposure determined the microbial response to heat stress. The corals that had lived in a more stable, cooler environment (i.e., the MV pool) bleached significantly and their microbiomes were significantly affected by the heat stress after 20 h (ANOSIM, R = 0.161, P < 0.001; Supplementary Fig. 1B). However, the microbiome of MV corals did not become more similar to those of naturally heat-stressed HV corals, as detected by SIMilarity PERcentage (SIMPER) analysis (Supplementary Data 4). In contrast, coral colonies that had lived in the warmer, more variable environment (i.e., the HV pool) bleached less and maintained their original microbial communities. They were more resistant to the heat stress and their microbiome showed no significant response after 5 and 20 h

compared with control corals (ANOSIM, R = 0.062, P > 0.05; Fig. 1d and Supplementary Fig. 1A).

**Microbial functional profiles change with thermal habitat**. To further understand whether coral acclimatization can be explained by microbial community shifts in our data, we elucidated the associated cellular processes underlying the distinct microbial communities of HV and MV corals. Among all functional traits identified using predictive metagenomic analysis, 128 functional traits and 28 proteins distinguished the microbial communities between both pools (Supplementary Data 5).

Functional profiles of microbial communities were distinct between HV and MV corals (Table 1). At large, HV coral microbiomes were characterized by enrichment of functions related to metabolism (Supplementary Data 5). At the level of proteins (Table 1), several functions related to carbohydrate metabolism were enriched in HV corals, such as multiple proteins of the sugar transport system, specifically three fructose transport proteins, one ribose transport protein, and a fucose mutarotase enzyme (Table 1). Other enriched proteins in HV coral microbiomes included the nitrogen-fixation protein NifW, the reactive oxygen species-scavenger ferredoxin, and the bacterial chaperonin GroES (Table 1). Only three functions were enriched in MV coral microbiomes, two proteins aiding the transfer of functional groups between molecules and a bacterial exonuclease (Table 1).

**Bacterial taxa as indicators of coral heat tolerance**. We then analysed our data for the presence of candidate indicator taxa[23] to identify bacterial OTUs that characterize the unchanged microbiomes of corals in the HV pool. The stable microbial

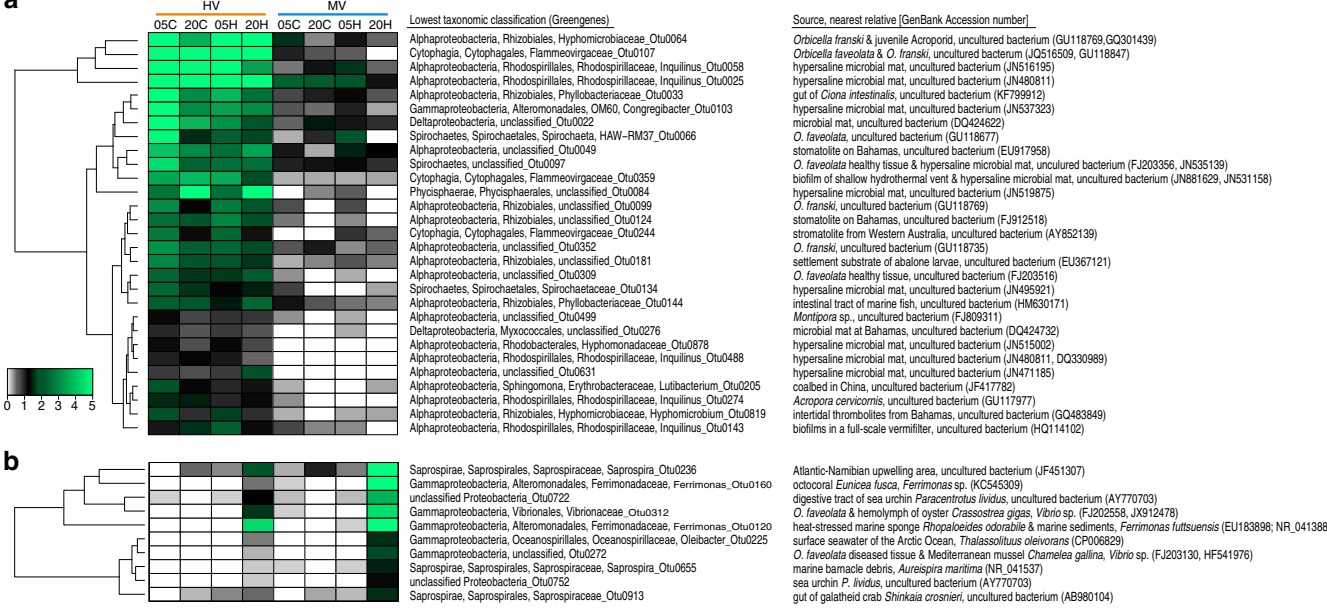

**Figure 2 | Bacterial indicator taxa of coral thermal environments and heat stress tolerance.** (**a**) Abundance of bacterial indicator taxa that characterize heat-tolerant corals in the HV pool over all treatments of the short-term heat stress experiment. (**b**) Abundance of bacterial indicator taxa that characterize the heat stress response in heat-sensitive corals in the MV pool. Each cell represents the square root transformed mean count of each indicator OTU per group ($n = 10$–13). Previous occurrences of identical or highly similar bacteria are listed with their environmental source and GenBank Accession number. HV, highly variable pool; MV, moderately variable pool; 05C and 20C = 5 and 20 h control (C) short-term heat stress experiment; 05H and 20H = 5 and 20 h treatment (H) short-term heat stress experiment.

community in the HV pool was characterized by a consistent set of microbial taxa across all control and heat treatments that differed from that of the less variable, cooler MV pool (Supplementary Data 3). About two-thirds of the indicator OTUs in the HV pool belonged to the class Alphaproteobacteria; the remaining OTUs belonged to the classes Cytophagia, Deltaproteobacteria, Phycisphaerae, Spirochaetes, and only one OTU belonged to the Gammaproteobacteria (Fig. 2a). Based on previous occurrences of these bacteria, most of the bacterial indicator taxa in HV corals were associated with saline environments and/or with other scleractinian corals (Fig. 2a and Supplementary Data 3). These OTUs were not (or only rarely) present in corals from the MV pool that displayed a response to heat stress. Further analyses focusing on the 20 h heat-stressed samples in comparison with all other treatment groups (i.e., 5 and 20 h controls and 5 h heat stress) revealed that microbial communities of corals from the HV pool had no distinct bacterial indicator taxa, while the same treatment group from the MV pool had 10 indicator OTUs that were characteristic of the heat-induced microbial shift in these more susceptible corals (Fig. 2b). The heat-stressed corals from the MV pool were characterized by members of the bacterial classes Gammaproteobacteria and Saprospirae, some of which were previously encountered in heat stressed or diseased tissues of marine organisms (Fig. 2b and Supplementary Data 3).

Importantly, and in line with identified bacterial indicator taxa of heat tolerance, the functional microbial profiles (see previous section) were also encoded by different bacterial classes. In HV corals, Alphaproteobacteria were responsible for a large fraction of the functional enrichment, with the family Rhodospirillaceae and the genus *Inquilinus* as prevalent contributors. In contrast, in MV corals Gammaproteobacteria were predominantly associated with enriched functional profiles. The main contributing families in the Gammaproteobacteria were Hahellaceae, Alteromonadaceae (strain BD2-13, genera *Alteromonas* and *Glaciecola*), and Vibrionaceae (genus *Vibrio*) (Supplementary Data 6).

**Discussion**

Host genetic adaptation[10] and differences in the *Symbiodinium* community composition[24] align with differences in thermal sensitivity in corals, yet the role of bacteria in coral thermal resilience is unknown. Studies in other systems argue for a role of bacteria in conferring heat tolerance[2] and disease resilience[1]. To begin to address this question for corals, we conducted a long-term reciprocal transplantation experiment of *A. hyacinthus* between two thermally distinct environments (i.e., a highly variable HV pool and a moderately variable MV pool) on Ofu Island, located in the U.S. National Park of American Samoa. Native coral colonies in the two pools had different microbial communities. Moreover, after 17 months of transplantation, microbiomes of non-native corals transplanted into each pool adjusted to the new environmental conditions and were not significantly different from native corals in the same pool. In a short-term heat stress experiment, the microbial community responded within 20 h for corals that had been transplanted to a more stable, cooler environment. But colonies living in the warmer, more variable environment for 17 months bleached less and maintained their microbial communities. The robust and stable microbiome of corals from the highly variable HV pool was characterized by a consistent set of microbial taxa across all control and heat treatments that were not (or only rarely) present in the susceptible corals transplanted to the more stable, cooler MV pool.

What remains to be determined is whether corals and microbes are solely responding to the same environmental changes, or whether differences in the bleaching response of the corals have a direct relationship to the differences in the associated

microbiomes. Experimental replacement of obligate symbionts from corals in the HV pool to corals in the MV pool with a subsequent demonstration of acquired heat tolerance would ultimately demonstrate the effect of host-associated bacteria on host ecology. This was recently shown for Aphids harbouring a single predominant bacterial strain[2], but these methods are currently unavailable for corals. Nevertheless, while we did not transfer microbiomes between HV and MV corals, the long-term transplantation effectively resulted in HV corals harbouring an MV microbiome and MV corals harbouring an HV microbiome associated with higher and lower susceptibility to coral bleaching, respectively.

Taxonomy-based functional profiling of the distinct microbial communities of HV and MV corals provided insight into the associated cellular processes underlying the differences in the coral heat stress response. Following the differences in microbial community composition, functional profiles were different between HV and MV corals, with an enrichment of several protein functions related to carbohydrate metabolism in HV coral microbiomes. These may represent signatures of bacterial heat tolerance, as indicated by the role of altered carbohydrate composition in the mucus of heat-stressed corals[25]. In this regard, an enzyme of Fucose, shown to undergo the largest increase of carbohydrates in coral mucus under heat stress[25], was enriched in HV corals from our study. We found three additional proteins that are part of the Fructose transport system to be enriched in HV corals. In support of these results, carbohydrate transport and metabolism (for example, the enzyme fructose-1-phosphate kinase) are under strong regulation in Streptococcus mutans to cope with heat stress[26]. Other enriched proteins that provide a link to increased thermal tolerance of bacteria in HV corals included the nitrogen-fixation protein NifW[27], ferredoxin, a scavenger of reactive oxygen species that increases heat tolerance in Chlamydomonas[28] and also in coral larvae[29,30], and the bacterial chaperonin GroES, which is a homologue to Hsp10 and assists in correct folding of proteins under heat stress[31]. These data support a functional restructuring of the microbial metabolic network in stress-resistant corals[32].

The identification of candidate bacterial indicator taxa that characterize the unchanged microbiome of heat-tolerant corals in the HV pool represents spatially explicit bacteria that are hypothesized to fulfil functional niches[21]. These bacteria remained stably associated even throughout the short-term heat stress experiment, and thus, they may contribute to the resilience of the coral holobiont to the challenging prevailing temperatures in the HV environment. In comparison, corals in the MV pool had no consistently distinct associated bacterial taxa. Rather, we found heat-associated changes in the microbiome of MV corals characterized by bacterial taxa that were previously recorded in diseased or stressed marine organisms.

Integrating the results of the different analyses, Alphaproteo-bacteria were responsible for a large fraction of the functional enrichment in the HV pool, with the family Rhodospirillaceae and the genus Inquilinus as prevalent contributors. This observation relates well with the significantly higher abundance of Rhodospirillaceae in the HV pool and the identification of indicator bacteria from the genus Inquilinus in the HV pool, and thus suggests a possible role of Rhodospirillaceae in the thermal tolerance of the corals. Encountering the bacterium Inquilinus in the heat-tolerant corals may seem unusual, because it has been initially isolated from the lungs of cystic fibrosis patients. So far, Inquilinus has very rarely been reported from other environments[33] and its putative role in corals requires experimental verification. Strikingly, however, despite the arguably pronounced differences between both habitats, the human lung and the coral host are both mucous environments and Inquilinus is

thermotolerant with a maximum cultivation temperature of up to 42 °C (ref. 34). Conversely, Gammaproteobacteria were associated with more than half of the bacterial functional traits of corals in the MV pool. The main contributing families in the Gammaproteobacteria were Hahellaceae, containing the well-known coral symbiont Endozoicomonas[19,35], and Alteromonadaceae, which was represented by strain BD2-13 and the genera Alteromonas and Glaciecola. Members of Glaciecola are better known for their occurrence in polar regions and the cold adaptation of their genomes[36], representing another link to the differences in thermal tolerance between the different environments of the HV and MV pools. A third family of Gammaproteobacteria that comprises many disease agents and sources of virulence for corals, the Vibrionaceae, further contributed to the functional profile of microbial communities in the MV pool. Among them was Vibrio shilonii, a bacterium that has been associated with coral bleaching in the coral Oculina patagonica[37,38], providing another possible connection to the higher incidence of bleaching of the MV corals during the heat stress experiment.

In the face of rapid climate change, long-lived sessile animals such as corals are considered particularly susceptible, and it is important to understand the mechanisms contributing to their resilience[39]. Beyond physiological acclimatization[18], host adaptation[10,18], and assisted migration of heat-tolerant alleles[40,41], microbial adaptation constitutes another possible mechanism to counteract the effects of environmental change. This idea is formulated in the coral probiotic hypothesis that posits a dynamic relationship between microorganisms and environmental conditions, which selects for the most advantageous coral holobiont[13]. Our data provide evidence for a community of microbes associated with coral thermal resilience patterns across variable habitats. In particular, our analyses from spatially close, but different thermal environments emphasize the flexibility of coral microbiomes, even within the same coral species. Our data demonstrate that differences in coral microbial communities are largely independent of the underlying coral genotype, but rather align with the prevailing environment as corroborated by our long-term transplantation experiment. Most importantly, microbiomes from a more stable cooler environment show a pronounced microbial change during bleaching conditions, while microbiomes from a highly variable warmer environment frequently exposed to heat stress maintain their original bacterial community composition throughout the heat stress. This result highlights that the response to heat stress differs with microbial community composition and potentially suggests a role of the bacterial community in the response of corals to heat stress. Although the current data do not allow us to discern whether these microbial community differences influence host thermal resilience or are influenced by it, the observed patterns align well with the notion that microbial adaptation may constitute a fast response mechanisms of corals to cope with environmental changes.

## Methods

**Sampling design and experimental heat stress experiment.** Ten colonies of A. hyacinthus (var. surculosa) from the south lagoon of Ofu Island, American Samoa (14°11'S, 169°36'W) were collected from each of two locations with different natural temperature regimes, an HV pool and an MV pool, in July 2011. The colonies were fragmented into 240 nubbins and distributed on 12 egg-crate platforms with the same 20 genotypes per platform, 10 genotypes from each pool environment[22]. Six of these platforms were transplanted into each pool. After 17 months of acclimatization to the different pool environments, a subset of visually healthy coral nubbins was used in a bleaching experiment in the laboratory.

In December 2012 (southern hemisphere summer), coral nubbins were transferred from the lagoon to six 6-l experimental tanks and the bleaching simulation was started immediately. Corals in the heat stress treatment were exposed to increasing temperatures from 29 to 35 °C over 3 h, held at 35 °C for 1 h, and returned to 29 °C within 2 h, while the control temperature was kept

at a constant 29 °C (Fig. 1b). Samples were collected after 5 h to capture the early onset of heat stress and again after 20 h to coincide with the onset of visually detected bleaching[22]. Four genetically identical nubbins were placed in each experimental tank. Two of these nubbins came from the HV pool and the other two from the MV pool. A nubbin coming from each pool was collected at the 5 h time point and later at the 20 h time point when visual signs of bleaching became apparent. Bleaching severity was scored visually for each genotype in relation to the control nubbins on a scale from 1 (normal) to 5 (completely bleached). Nubbins were fragmented into two pieces; one half was stored in RNA fixing solution for molecular analysis (and subsequently frozen at −80 °C) and the other half in 100% ethanol for pigment quantification. The experiments were repeated for a total of 19 genotypes over 4 days. The light intensity in the tanks was approx. 700 µmol m$^{-2}$ s$^{-1}$ (Apogee Quantum meter MQ-200) and water flow through tanks was kept at 5 l h$^{-1}$ with unfiltered seawater from the lagoon.

**RNA extraction.** Coral fragments were defrosted, excess liquid was removed, and a small fragment was transferred into a 1.5-ml tube containing 100 µl of 0.3 mm ceramic beads and 1 ml of TRIzol reagent (ThermoFisher Scientific). Coral tissue was disrupted on a Vortex Genie (Scientific Industries) for 5 min, incubated at room temperature for 5 min, then 200 µl of chloroform was added, and the tube was inverted 15 times. After a 3-min incubation at room temperature, phase separation was completed through centrifugation at 12,000g for 15 min at 4 °C. The top aqueous phase, approximately 500 µl was then transferred to a new tube and mixed with 250 µl of 100% isopropanol and 250 µl of a high salt buffer (0.8 M sodium citrate and 1.2 M sodium chloride). Total RNA was precipitated at 4 °C overnight and pelleted through centrifugation at 12,000g for 10 min at 4 °C. The supernatant was discarded; the RNA pellet was washed in 1 ml of 75% ethanol by gentle pipetting and centrifuged again at 7,500g for 5 min at 4 °C. The supernatant was discarded, the pellet was dried at room temperature and resuspended in 30 µl of RNase-free water.

**PCR amplification and sequencing conditions.** Isolated total RNA was reverse transcribed using the SuperScript II reverse transcriptase according to the manufacturer's protocol (Invitrogen, Carlsbad, USA). 16S rRNA genes from cDNA were amplified using the primers 784F and 1061R (ref. 42) with MiSeq 16S adapter sequences (forward: 5′-TCGTCGGCAGCGTCAGATGTGTATAAGAGACAGAG GATTAGATACCCTGGTA-3′; reverse: 5′-GTCTCGTGGGCTCGGAGATGTGT ATAAGAGACAGCRRCACGAGCTGACGAC-3′; Illumina overhang adaptor sequences are underlined). Twenty µl PCRs were run in triplicate per sample using Qiagen multiplex PCR master mix, 2 µl cDNA as a template and a final primer concentration of 0.25 µM. PCR cycling conditions were 95 °C for 15 min, followed by 27 cycles of 95 °C for 30 s, 55 °C for 30 s and 72 °C for 30 s, with a final extension time of 10 min. Triplicate PCRs were pooled and successful amplification was visualized on the Bioanalyzer (Agilent Technologies, Santa Clara, USA). MiSeq indexing adaptors were added via PCR according to the Illumina 16S metagenomic sequencing library preparation protocol. PCR products were run on a 2% ultra-pure agarose gel (Ultrapure Agarose, Life Technologies) and purified using the Zymoclean DNA large fragment recovery kit (Zymo Research, Irvine, USA). 16S rRNA gene amplicon libraries were sequenced at the KAUST sequencing facility on the Illumina MiSeq platform using 2 × 300 bp overlapping paired-end reads with a 10% phiX control.

**Sequence data processing.** Error correction, taxonomical classification, and alpha and beta diversity indices were processed with mothur v.1.33.3 (ref. 43). Briefly, sequence reads were trimmed and joined into contigs. Contigs longer than 315 bp (2.5% of all reads) and those with ambiguously called bases were excluded from the analysis. Unique contigs that were identified exactly one time over all samples were removed from further analyses. SILVA reference database was used for 16S rRNA gene amplicon alignment. Sequences were pre-clustered, allowing for up to a 2 nt difference between the sequences. Chimeras were identified and removed using the UCHIME[44] function as implemented in mothur. Sequences were then classified using the Greengenes database with a 60% bootstrap cutoff. Only sequences that were classified as deriving from bacteria were kept and subsampled to 4,286, the lowest number of sequences over all samples. For further analyses, subsampled sequences were clustered into Operational Taxonomic Units (OTUs) at a 97% similarity cutoff and reference sequences for each OTU were determined (Supplementary Data 1).

**16S rRNA gene-based microbial community analysis.** Phylogenetically annotated 16S sequences (see above) were used to create bacterial community composition stacked column plots at the family level using the means of relative abundances from replicated samples of reciprocally transplanted corals at 5 and 20 h time points during the short-term heat stress experiment (Fig. 1d). The linear discriminant analysis (LDA) effect size (LEfSe) method[45] was used to test for bacterial families that were significantly different in their abundance in corals transplanted to the HV and MV pool, respectively (LDA > 2.0).

Differences in microbial communities were tested using ANOSIM with 9,999 permutations. First, the effect of the reciprocal transplantation to thermally distinct pools (i.e., HV and MV) on the microbial communities was investigated. Because the microbiomes of non-native corals transplanted into each pool

were indistinguishable from the microbiomes of the native corals in the same pool and cross-transplanted fragments were significantly different from their back-transplanted (genet) counterparts, we grouped coral fragments by pool of destination for all following analyses. Next, the differences in microbial communities during the short-term heat stress experiment were tested over all samples and separately for each pool of destination (i.e., the HV and MV pools; Supplementary Information Fig. 1). Results from the ANOSIM were visualized in non-metric multidimensional scaling ordination plots with ellipses drawn around each group's centroid using the package 'ggplot'[46] as implemented in R software v 3.1.3 (ref. 47). SIMilarity PERcentage (SIMPER) analysis was conducted to test whether microbial communities in heat-stressed corals from the MV pool became more similar to microbial communities from HV corals. All multivariate tests were performed on Bray Curtis distances of log $(x + 1)$ transformed OTU counts using PRIMER-E v6 software package.

The software mothur was further used to obtain shared microbial OTUs (command 'get.coremicrobiome') across all *A. hyacinthus* corals and across back-transplanted and cross-transplanted corals (Supplementary Data 2).

**Bacterial indicator species representative of HV and MV corals.** The statistical package IndicSpecies[23] is commonly applied to analyse the strength and statistical significance of the relationship between species occurrence and/or abundance and groups of sites. In this study, we employed IndicSpecies as implemented in R to identify OTUs that were significantly associated with corals in the HV and MV pools, respectively, and with corals from these pools after the short-term heat stress experiment. The analysis was conducted on OTU counts (Supplementary Data 1) excluding OTUs < 20 reads. All samples were assigned to their pool of destination (HV or MV) and to one of four treatment groups (05C, 05H, 20C, 20H) using the command 'groups'. IndicSpecies was run using the command 'multipatt' with 999 permutations. Significant OTUs were summarized (command 'summary') for each group separately and for all combinations thereof. Significant OTUs were false discovery rate corrected (10%) following Benjamini & Hochberg[48] and the representative sequence of each significant indicator OTU was then BLASTed against GenBank nr database to identify previous occurrences of identical or highly similar bacteria.

**Co-variation of coral host genotypes and bacterial symbionts.** We screened for the presence of distinct bacterial taxa that are associated with distinct coral colonies (genotypes) by querying the OTU count table (Supplementary Data 1) for bacterial OTUs that were exclusively present in all samples of a given coral genotype and absent from all other genotype samples. To statistically test for exclusive coral host genotype–bacterial taxon pairings, we tested for coral colony-specific bacterial indicator species using IndicSpecies (use of IndicSpecies as detailed above).

**Functional profiling based on bacterial taxonomy.** We applied Phylogenetic Investigation of Communities by Reconstruction of Unobserved States (PICRUSt) to predict metagenomic functional content from the 16S rRNA marker gene[49]. To account for differences in gene copy number, the command 'normalize_by_copy_number.py' was applied to the OTU abundance table. Metagenome predictions were conducted using 'predict_metagenomes.py' and individual KEGG Orthology groups (KOs) were summarized at KEGG-Pathway level 1, 2, and 3 with 'categorize_by_function.py'. For quality control, weighted Nearest Sequenced Taxon Index (weighted NSTI) was calculated for each sample and the NSTI was found to be in a satisfactory range for metagenomic predictions (mean NSTI = 0.12 ± 0.04 s.d.)[49]. The count table of metagenome predictions was further analysed using the Galaxy web application (https://huttenhower.sph.harvard.edu/galaxy/) and the LEfSe method[45] to identify significantly different metagenome functions of microbial communities between the HV and MV pools (LDA > 2.0 for levels 1–3, LDA > 1.5 for individual KOs). Metagenome functional contributions were partitioned to each OTU using 'metagenome_contributions.py'. This analysis yielded an absolute numerical count of the contribution to each KO function for each OTU in each sample. Normalized metagenome contributions of KOs were summarized per significant level 3 KEGG-Pathway (see above) for each OTU to derive bacterial class level contributions per pathway (Supplementary Data 6).

**Data availability.** Sequence data determined in this study are available at NCBI under BioProject Accession PRJNA319637 (https://www.ncbi.nlm.nih.gov/bioproject/PRJNA319637/). OTU reference sequences are available under GenBank Accession numbers KY373275-KY377940. Other data are available in the Supplementary Data files.

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

## Acknowledgements

We thank Craig Michell (KAUST) for sequence library preparation, Yi Jin Liew (KAUST) and Sebastian Steinke (KAUST) for assistance with OTU mapping to metagenome function, Shobhit Agrawal (KAUST) for support with statistical analysis, Ivan Gromicho (KAUST) for implementing the concepts for Fig. 1 and KAUST Bioscience Core Lab for sequencing. Research reported in this publication was supported by baseline research funds to C.R.V. and Red Sea Research Center funded projects FCC/1/1973-10-01 and FCC/1/1973-18-01 by KAUST.

## Author contributions

S.R.P., C.R.V. and F.O.S. designed and conceived the experiment. F.O.S. and L.K.Y. generated data; M.Z. analysed data; M.Z., C.R.V. and S.R.P. interpreted data; and M.Z. and C.R.V. wrote the manuscript. All authors contributed to and approved the manuscript.

## Additional information

**Competing financial interests:** The authors declare no competing financial interests.

