## [Peer Review File · Nature Communications]

PEER REVIEW FILE

Reviewers' Comments:

Reviewer #1 (Remarks to the Author):

As I said in my previous review of this manuscript for Nature Climate Change, the paper presents several very exciting and important findings elucidating ecological interactions between corals and microbes. I copy-pasted my original (unchanged) impressions of what is most exciting about this paper in the end of this document.

The authors carefully reworked the text and analyses according to most of my recommendations, and now I am mostly happy with it. Current version of Fig. 2 is just awesome.

There is one notable exception, though: I still think that analysis of functional profiles of microbial communities as illustrated on Fig. 3 is meaningless - or at least, meaningless without a more extended discussion. The functions claimed to be over- or under-represented among microbiomes are all basic and essential, without them no microbe can live. Note that this is not gene expression – this is gene content in bacterial genomes – so how can one bacterium contain significantly less replication or translation genes than another? And what does it mean biologically? Note that I do not contend the general methodology of PICRUSt analysis (although I am skeptical about it) or way this analysis was performed in this case, so the response to my earlier similar comment in the rebuttal letter is largely beside the point.

One possibility to explain Fig.3's perplexing result is that increase in number and diversity of metabolic genes (which is at least conceivable) in the HV pool brings about relative decline in proportion of information-processing genes, simply numerically: as the larger fraction of all

genes is taken up by the metabolism, other genes look like they are declining in proportion even though their absolute number does not change. Is this what is happening?

In any case, if the authors insist on keeping Fig. 3, they must add a paragraph to discussion explaining really well what the differential distribution of those basic and essential biological functions really might mean. Currently all we have are cursory statements that more informational genes in MV pool might indicate bacterial stress response and more metabolic genes in HV pool might indicate higher stress tolerance. I see no logic and no references to support these statements.

I strongly suggest removing Fig. 3 and restricting the functional microbiome analysis to Table 1 and discussion about genes listed there (which is pretty good).

Minor, suggestive things:

The authors' response to my first [positive] comment – “Hurray! there ain't no such thing as a hologenome”, see quote below – is mostly beside the point. They note that there are taxa that are always associated with these corals – that's OK but not relevant since coadaptation and coevolution and general “hologenome-ness” is all about co-variation (or absence thereof, as seems to be the case here) of host and symbiont genetics (in this case across environments). When there is no variation (i.e., bugs are found in all corals), there is no co-variation either. So identifying “core” bacteria is not helpful to address the “hologenome” issue. I suggest including a sentence in discussion about “hologenome” controversy and pointing out the lack of covariance between coral genet and either overall microbiome composition or specific OTUs (or maybe some of them do stay true to their host?.. can such OTUs be found? it would be very interesting)

Discussion has a dedicated section “Bacterial taxa as indicators of coral thermal resilience” but then the next section, supposedly about genes/proteins (Functional profiles of coral microbiomes are distinct between thermally different environments) again starts talking about differences in bacterial taxa. I suggest sorting genes and bacterial taxa into two separate discussion sections.

Supp. Fig 1 is not exactly what I thought of... The advantage of DAPC (the method I suggested) over other multivariate approaches is that its eigenvectors are calculated to maximize not the

total explained variance but the difference of interest – in this case, it would be difference between native HV and MV pool microbiomes. More importantly, in DAPC one can quantitatively measure to which extent new samples (microbiomes after transplantation or short-term heat stress) become similar to the original groups – the method would report percentage of new samples that are classified as belonging to one of the original groups.

That said, of course I do not insist that the authors use DAPC. Still, I would be happier if the authors modified the NMDS plots to make them a little more explanatory (in the present forms it is really hard to see what is going on in them) – for example draw labeled ellipses (function `ordiellipse` of `vegan` package in R) showing the extent and position of represented groups?

Finally, a general advice to the authors about how to present their revised material. Next time, please (i) number lines in the main manuscript; (ii) paste the relevant edited passages from the text right into the “response to reviewers” document. Just mentioning that changes are there is not quite enough – I have to scroll through the manuscript every time and hope that what I find is indeed all that was changed. This is inefficient and time-consuming.

“Notorious” *Endozoicomonas*: I just want to make sure that the authors really want to imply that this bug is “famous for being bad” (which is what “notorious” means) – I don’t think there is any inherent “badness” to it, no?

#-----

Relevant content from my earlier review (just in case):

There are three results in this manuscript that I find extremely interesting:

1. The most interesting one: that bacterial community is fully dependent on the environment, and not at all on coral genotype. This is an extremely valuable observation, implying that there is no co-evolution of corals and microbes at the intra-specific level - otherwise there would not be such a complete change of microbiome depending on the environment. To show this with numbers, it might be possible to show partitioning of microbiome variance into environmental, genotypic, and unexplained components. So we can say that "hologenome" concept (host

genome + microbial genomes = single unit of evolution) does not apply here, which would clarify things for many researchers as well as for funding agencies (no, we don't have to consider microbiome when studying genetics of coral adaptation; microbes are just another part of coral reef ecosystem and should be addressed in ecological, not genetic, terms).

2. That both corals and their microbiomes become insensitive to temperature stress if they spend 17 months in a high-stress environment. Corals acclimatize, as has been shown before, and microbiome gets replaced by a more resistant local community. It is important to note that based on just this data it is not possible to claim causal connection between the two: it remains unclear whether microbiome replacement affects coral acclimatization in any way (as the abstract and passages in the Discussion correctly state).

3. In corals from low-stress environment, heat stress leads to change in microbiome that does not make it more similar to heat-resistant microbiome. This is not too surprising but still a very valuable fact to establish: the switch to heat-resistant microbiome cannot be accomplished in a single acute heat stress event.

Reviewer #3 (Remarks to the Author):

This is timely and valuable research given the recent worldwide bleaching event that is still impacting coral reefs. The insights of this work are highly valuable to the coral reef research community at this time and I highly recommend that this work be published and available to the community as soon as possible.

The application of the Caceres et al 2009 indices and statistical inference, is excellent and a more detailed outline in the methods section would aid other microbial ecologists in its use. The application of this approach also highlights the value of considering ecological principals for microbial analyses particularly given the difficulty is differentiating patterns microbiome and

causation. The authors note these competing hypotheses in the studies summary, which is an important and often overlooked in the literature. The authors have also successfully addressed the competing hypothesis that exist in coral microbiome literature and have elegantly avoided biased interpretations of the data within this study. In doing so the authors present a well balanced and very insightful study.

The authors have adequately addresses all concerns raised by the two reviewers whilst the under consideration by Nature Climate Change.

Specific Comments:

Figure 1a and 1b are difficult to interpret. I would recommend that the sizing be similar to that of 1C. Similarly Figure 3 would benefit from larger pie charts for the reader to easily interpret not just the more obvious differences but also interpret the finer details. I would like to have seen the SI figure (nMDS) in the main text of the manuscript and I would suggest incorporating it as figure 1C while using the phylogenetic bar charts as 1D.

I have no further recommendation in regards to the content, analyses or compilation of the study.

The research is well suited to Nature Communications and in light of the current bleaching events worldwide, and subsequent research efforts to understand their long-term impact, will become a highly cited work.

Reviewer #1:

As I said in my previous review of this manuscript for Nature Climate Change, the paper presents several very exciting and important findings elucidating ecological interactions between corals and microbes. I copy-pasted my original (unchanged) impressions of what is most exciting about this paper in the end of this document.

The authors carefully reworked the text and analyses according to most of my recommendations, and now I am mostly happy with it. Current version of Fig. 2 is just awesome.

We thank the reviewer for his/her support and excitement. We are particularly excited about this work too.

There is one notable exception, though: I still think that analysis of functional profiles of microbial communities as illustrated on Fig. 3 is meaningless - or at least, meaningless without a more extended discussion. The functions claimed to be over- or under-represented among microbiomes are all basic and essential, without them no microbe can live. Note that this is not gene expression – this is gene content in bacterial genomes – so how can one bacterium contain significantly less replication or translation genes than another? And what does it mean biologically? Note that I do not contend the general methodology of PICRUSt analysis (although I am skeptical about it) or way this analysis was performed in this case, so the response to my earlier similar comment in the rebuttal letter is largely beside the point.

One possibility to explain Fig.3's perplexing result is that increase in number and diversity of metabolic genes (which is at least conceivable) in the HV pool brings about relative decline in proportion of information-processing genes, simply numerically: as the larger fraction of all genes is taken up by the metabolism, other genes look like they are declining in proportion even though their absolute number does not change. Is this what is happening?

We thank the reviewer for his/her comment regarding results from our PICRUSt analyses. Indeed, enrichment of protein functions must refer to an increase of gene content in bacterial genomes, rather than a mere over- or under-expression of genes.

To check for a numerical increase/decrease of bacterial function (taken over by an respective increase/decrease of other bacterial functions), we checked the absolute counts of metagenome functions in the untransformed dataset and calculated percent contributions from at-large categories (i.e., metabolism, genetic information processing, etc.).

We found that, generally, counts for metabolism-associated functions were slightly increased in HV corals, whereas counts for genetic information processing were slightly decreased in HV corals compared to MV corals. Indeed, this might have led to higher proportions of metabolism functions and lower proportions of genetic information processing in the HV samples, and could provide an explanation for the patterns observed. Accordingly, we agree with the reviewer's assessment and have followed his recommendations to remove Figure 3 (further detailed below).

In any case, if the authors insist on keeping Fig. 3, they must add a paragraph to discussion explaining really well what the differential distribution of those basic and essential biological functions really might mean. Currently all we have are cursory statements that more informational genes in MV pool might indicate bacterial stress response and more metabolic genes in HV pool might indicate higher stress tolerance. I see no logic and no references to support these statements.

In addition to our response above, we agree that the current data do not allow us to unequivocally determine where the differences come from (i.e., simply numerical differences or biological/bacterial community differences). Following the reviewer's notion and based on editorial recommendation, we removed Figure 3 from the revised manuscript. We also removed respective passages in the revised manuscript that infer such functional differences, such as the statement that enrichment of information-processing genes in MV pool might indicate bacterial stress response and enrichment of metabolic genes in HV pool might indicate higher stress tolerance.

I strongly suggest removing Fig. 3 and restricting the functional microbiome analysis to Table 1 and discussion about genes listed there (which is pretty good).

Thank you. We too think that differences based on actual proteins provide a more straightforward and meaningful way of comparing HV and MV corals. As pointed out above, we have removed Figure 3 from the revised manuscript and lay the focus of the functional analyses on results from Table 1.

Minor, suggestive things:

The authors' response to my first [positive] comment – “Hurray! there ain't no such thing as a hologenome”, see quote below – is mostly beside the point. They note that there are taxa that are always associated with these corals – that's OK but not relevant since coadaptation and coevolution and general “hologenome-ness” is all about co-variation (or absence thereof, as seems to be the case here) of host and symbiont genetics (in this case across environments). When there is no variation (i.e., bugs are found in all corals), there is no co-variation either. So identifying “core” bacteria is not helpful to address the “hologenome” issue. I suggest including a sentence in discussion about “hologenome” controversy and pointing out the lack of covariance between coral genet and either overall microbiome composition or specific OTUs (or maybe some of them do stay true to their host?.. can such OTUs be found? it would be very interesting)

Valid point. Prompted by the reviewers' suggestion we further interrogated our data for covariance of genotypes and OTUs. For these analyses, we considered samples from 14 coral genotypes that had at least 4 replicates in the data set (regardless of pools and/or treatments).

Results are as follows:

1. Variance across genotypes:

Of the total 4,801 OTUs, 94 OTUs were present in all samples from several or all genotypes (including the 'core microbes' present in all samples). Of these 94 OTUs, only 39 were consistently present in all samples of a single given genotype without being present in all samples from another genotype. Following the reviewer's notion, we did not find a single OTU that was exclusive to a genotype – if an OTU was present in all samples of a given genotype, it always also occurred in other genotype samples (although not in all samples of these).

2. Variance within genotypes:

We further checked for the presence of OTUs that occurred exclusively in only one genotype – but not in all samples of that genotype (with a minimum read number equal to the minimum number of samples per genotype group = 4). 272 OTUs were exclusive to one genotype, but not consistently present across all samples of that genotype. The mean sum of reads for these OTUs was 12.7, the median 7. With such low numbers of reads in the OTUs, the issue of false negatives arises – i.e. an OTU not being detected, despite its presence in a sample.

To further address the issue statistically (-> stochasticity of genotype–OTU pairings), we screened our data for Indicator Species (IndicSpecies analysis) of genotypes, applying the same approach that was used on pool and treatment differences in the manuscript. We found no significant association between a given genotype and a bacterial OTU (even before FDR correction).

Thus, we agree with the reviewer and, following the reviewer's recommendation, we added a respective passage to the discussion section that highlights the controversy of the hologenome theory.

Corresponding section in revised manuscript reads: "To test for the presence of co-variation of coral host genotypes and bacterial symbionts across environments that would point towards a relationship between host genetic variance and bacterial association, we tested whether distinct genotypes of *A. hyacinthus* were associated with distinct bacterial OTUs. We did not find any particular bacterial taxon associated with any particular coral colony (genotype), and hence, our study indicates a lack of covariance between coral genotype and microbiome composition. The apparent lack of covariance in *A. hyacinthus* requires further attention and argues against a heritable microbial component of the coral holobiont. Our data may thus contribute important insight to the debate around the hologenome theory and further studies are warranted that investigate in detail the relationship between animal or plant hosts and associated bacterial symbionts (Theis, Dheilly et al. 2016)."

The additional analyses outlined above are described in a new dedicated methods section that reads: **Co-variation of coral host genotypes and bacterial symbionts** - We screened for the presence of distinct bacterial taxa that are associated with distinct coral colonies (genotypes) by querying the OTU

count table (Supplementary Data 1) for bacterial OTUs that were exclusively present in all samples of a given coral genotype and absent from all other genotype samples. To statistically test for exclusive coral host genotype-bacterial taxon- pairings, we tested for coral colony-specific bacterial indicator species using IndicSpecies (use of IndicSpecies as detailed above)."

Discussion has a dedicated section "Bacterial taxa as indicators of coral thermal resilience" but then the next section, supposedly about genes/proteins (Functional profiles of coral microbiomes are distinct between thermally different environments) again starts talking about differences in bacterial taxa. I suggest sorting genes and bacterial taxa into two separate discussion sections.

Following the reviewer's suggestion, we have restructured these sections to improve the flow of the manuscript:

1. The section: "Bacterial community dynamics are linked to patterns of coral heat tolerance" is now directly followed by the section on functional analyses: "Functional profiles of coral microbiomes are distinct between thermally different environments".

2. The section on Indicator taxa: "Bacterial taxa as indicators of coral thermal resilience" is moved after the functional analysis.

3. The paragraphs on metagenome contributions of distinct bacterial taxa (previously directly after functional profiling) now follow the section on Indicator taxa and we added a separate heading: "Bacterial taxonomic and functional profiles are linked to coral thermal resilience"

Supp. Fig 1 is not exactly what I thought of... The advantage of DAPC (the method I suggested) over other multivariate approaches is that its eigenvectors are calculated to maximize not the total explained variance but the difference of interest – in this case, it would be difference between native HV and MV pool microbiomes. More importantly, in DAPC one can quantitatively measure to which extent new samples (microbiomes after transplantation or short-term heat stress) become similar to the original groups – the method would report percentage of new samples that are classified as belonging to one of the original groups.

That said, of course I do not insist that the authors use DAPC. Still, I would be happier if the authors modified the NMDS plots to make them a little more explanatory (in the present forms it is really hard to see what is going on in them) – for example draw labeled ellipses (function ordiellipse of vegan package in R) showing the extent and position of represented groups?

We appreciate the reviewer's suggestion regarding data visualization and analysis. As suggested, we re-analyzed the data using DAPC and also found a good separation of microbiomes after the transplantation (see figure below) using this alternative method. However, we prefer to use the nMDS plots as they directly visualize the results from our statistical routines (i.e. ANOSIM, etc.).

Following the reviewer's suggestion, in the revised manuscript we have added ellipses around the group centroids to the nMDS plots to improve clarity (see Figure 1C and Supplemental Figure 1).

Figure: DAPC analysis of coral microbiomes by pool of destination (blue = HV, red = MV). The two groups are well separated along discriminant function 1 (note that this is a density plot over only 1 discriminant function).

Finally, a general advice to the authors about how to present their revised material. Next time, please (i) number lines in the main manuscript; (ii) paste the relevant edited passages from the text right into the "response to reviewers" document. Just mentioning that changes are there is not quite enough – I have to scroll through the manuscript every time and hope that what I find is indeed all that was changed. This is inefficient and time-consuming.

Noted and apologies. Indeed, this is inconvenient. We thought we had provided a manuscript version with tracked changes for our last submission to avoid this. For this revision, we have copied all revised manuscript passages into the replies to the reviewer's comments.

"Notorious" Endozoicomonas: I just want to make sure that the authors really want to imply that this bug is "famous for being bad" (which is what "notorious" means) – I don't think there is any inherent "badness" to it, no?

We used the word "notorious" as in "well-known", but yes, the reviewer is right, this might be stretching it. We have replaced the word "notorious" with "well-known", because other readers will also likely interpret the phrase as pointed out by the reviewer.

The sentence now reads: "*The main contributing families in the Gammaproteobacteria were Endozoicomonaceae, containing the well-known coral symbiont Endozoicomonas*"

#-----

Relevant content from my earlier review (just in case):

There are three results in this manuscript that I find extremely interesting:

1. The most interesting one: that bacterial community is fully dependent on the environment, and not at all on coral genotype. This is an extremely valuable observation, implying that there is no co-evolution of corals and microbes at the intra-specific level - otherwise there would not be such a complete change of microbiome depending on the environment. To show this with numbers, it might be possible to show partitioning of microbiome variance into environmental, genotypic, and unexplained components. So we can say that "hologenome" concept (host genome + microbial genomes = single unit of evolution) does not apply here, which would clarify things for many researchers as well as for funding agencies (no, we don't have to consider microbiome when studying genetics of coral adaptation; microbes are just another part of coral reef ecosystem and should be addressed in ecological, not genetic, terms).
2. That both corals and their microbiomes become insensitive to temperature stress if they spend 17 months in a high-stress environment. Corals acclimatize, as has been shown before, and microbiome gets replaced by a more resistant local community. It is important to note that based on just this data it is not possible to claim causal connection between the two: it remains unclear whether microbiome replacement affects coral acclimatization in any way (as the abstract and passages in the Discussion correctly state).
3. In corals from low-stress environment, heat stress leads to change in microbiome that does not make it more similar to heat-resistant microbiome. This is not too surprising but still a very valuable fact to establish: the switch to heat-resistant microbiome cannot be accomplished in a single acute heat stress event.

Reviewer #3:

This is timely and valuable research given the recent worldwide bleaching event that is still impacting coral reefs. The insights of this work are highly valuable to the coral reef research community at this time and I highly recommend that this work be published and available to the community as soon as possible.

We thank the reviewer very much for his/her encouraging and supportive assessment.

The application of the Caceres et al 2009 indices and statistical inference, is excellent and a more detailed outline in the methods section would aid other microbial ecologists in its use. The application of this approach also highlights the value of considering ecological principals for microbial analyses particularly given the difficulty is differentiating patterns microbiome and causation. The authors note these competing hypotheses in the studies summary, which is an important and often overlooked in the literature. The authors have also successfully addressed the competing hypothesis that exist in coral microbiome literature and have elegantly avoided biased interpretations of the data within this study. In doing so the authors present a well balanced and very insightful study.

We thank the reviewer for the positive assessment of our work.

Following the reviewer's suggestion, we have extended the Materials and Methods section detailing the IndicSpecies analysis. Extended passages are underlined and the updated section in Materials and Methods now reads:

"The statistical package IndicSpecies is commonly applied to analyze the strength and statistical significance of the relationship between species occurrence and/or abundance and groups of sites. In this study, we employed IndicSpecies as implemented in R to identify OTUs that were significantly associated with corals in the HV and MV pool, respectively, and with corals from these pools after the short-term heat stress experiment. The analysis was conducted on OTU count data (Supplementary Data S1) excluding OTUs < 20 reads. All samples were assigned to their pool of destination (HV or MV) and to one of four treatment groups (05C, 05H, 20C, 20H) using the command 'groups'. IndicSpecies was run using the command 'multipatt' with 999 permutations. Significant OTUs were summarized (command 'summary') for each group separately and for all combinations thereof. Significant OTUs were FDR-corrected (10 %) following Benjamini and Hochberg and the representative sequence of each significant bacterial indicator OTU was then BLASTed against GenBank nr database to identify previous occurrences of identical or highly similar bacteria."

The authors have adequately addresses all concerns raised by the two reviewers whilst the under consideration by Nature Climate Change.

Specific Comments:

Figure 1a and 1b are difficult to interpret. I would recommend that the sizing be similar to that of 1C.

We have revised Figure 1 based on this recommendations of the reviewer and the editor. Figure 1A and 1B were increased in size and rearranged vertically to facilitate interpretation.

Similarly Figure 3 would benefit from larger pie charts for the reader to easily interpret not just the more obvious differences but also interpret the finer details.

Following the recommendations of reviewer 1 and the editor, we have removed Figure 3 from the manuscript. Information on metagenome contributions of different taxa is now/still available in the Supplement (Supplemental Table S6).

I would like to have seen the SI figure (nMDS) in the main text of the manuscript and I would suggest incorporating it as figure 1C while using the phylogenetic bar charts as 1D.

We thank the reviewer for this excellent suggestion. Figure 1 was revised and now includes the nMDS ordination (Fig 1C) to illustrate differences of microbial community composition between pools.

Based on a comment from Reviewer 1, we have improved the visualization of the nMDS plot to include ellipses around the group centroids to maximize the visibility of between group differences. Remaining MDS plots (for the heat stress experiment) are still available in the Supplement (Supplementary Information 1).

Following these changes, the phylogenetic bar chart is now designated as Figure 1D.

I have no further recommendation in regards to the content, analyses or compilation of the study. The research is well suited to Nature Communications and in light of the current bleaching events worldwide, and subsequent research efforts to understand their long-term impact, will become a highly cited work.

Thanks again!

Reviewers' Comments:

Reviewer #1 (Remarks to the Author):

Perfect! All my comments have been very carefully addressed now.

Bacterial community dynamics are linked to patterns of coral heat tolerance
Manuscript ID: NCOMMS-16-20499A

Our responses to the reviewer's comments are in **bold**.

Reviewer #1 (Remarks to the Author):

Perfect! All my comments have been very carefully addressed now.

We thank the reviewer for his/her support and approval. We are particularly excited about this work and happy that all comments were addressed to the reviewer's satisfaction.